# Inhibition of Biogenic Amines in Fermented Tilapia Surimi by Collaborative Fermentation of *Latilactobacillus sakei* and *Pediococcus acidilactici*

**DOI:** 10.3390/foods13203297

**Published:** 2024-10-17

**Authors:** Chunsheng Li, Chunhui Wang, Jianchao Deng, Di Wang, Hui Huang, Yongqiang Zhao, Shengjun Chen

**Affiliations:** 1Key Laboratory of Aquatic Product Processing, Ministry of Agriculture and Rural Affairs, National R&D Center for Aquatic Product Processing, South China Sea Fisheries Research Institute, Chinese Academy of Fishery Sciences, Guangzhou 510300, China; lichunsheng@scsfri.ac.cn (C.L.); chw0201@163.com (C.W.); dengjianchao@scsfri.ac.cn (J.D.); wangdi@scsfri.ac.cn (D.W.); huanghuigd@aliyun.com (H.H.); 2Key Laboratory of Efficient Utilization and Processing of Marine Fishery Resources of Hainan Province, Sanya Tropical Fisheries Research Institute, Sanya 572018, China

**Keywords:** tilapia surimi, cooperative fermentation, microbial community, biogenic amine, correlation network, *Latilactobacillus sakei*, *Pediococcus acidilactici*

## Abstract

Fermentation is an effective method for ameliorating the gelation properties of freshwater fish surimi, but the formation of biogenic amines (BAs) during fermentation should also be controlled. In this study, the BAs in fermented tilapia surimi were inhibited by the collaborative fermentation of *Latilactobacillus sakei* and *Pediococcus acidilactici*, followed by the revelation of the BA inhibition mechanism. Most of the BAs, and the total BA, as well as their precusor free amino acids (FAAs), were significantly reduced, while the umami FAAs, including glutamic acid and aspartic acid, were significantly enhanced after cooperative fermentation with starters. The high-throughput sequencing found that the spoilage microorganisms such as *Acinetobacter*, *Micrococcus*, and *Streptococcus* as well as *Pediococcus* were significantly inhibited, while *Latilactobacillus* rapidly became the dominant genus after cooperative fermentation, suggesting the better environment adaptability of *L. sakei* than *P. acidilactici*. The group-dimension correlation analysis suggested that *Lactiplantibacillus* had the greatest influence on the decrease in BAss. The quick acidification of starters, especially *L. sakei*, could inhibit the growth and metabolism of spoilage microorganisms to reduce BAs. *L. sakei* and *P. acidilactici* can be developed as the special starters to control the BA production in the fermented tilapia surimi through collaborative fermentation.

## 1. Introduction

Surimi products are famous worldwide because of their special mouthfeel from the gelation property [1]. In order to form a good gel, the surimi products are mainly produced from sea fish, due to their better gel strength than freshwater fish. However, as the demand for surimi products increases, more freshwater fish are used as raw material, and need to improve the gelation property [2]. As one of the main freshwater fish worldwide, tilapia are usually processed into filets [3,4], resulting in a low added value. The low gelation properties limit the processing of tilapia into surimi products. The addition of additives, such as protease inhibitors [5], microbial transglutaminase [6], and polysaccharides [7] is a traditional method to improve the gelation properties of surimi. Fermentation through lactic acid bacteria (LAB) is an effective method to enhance the gelation properties of freshwater fish surimi through acidification [8]. Previous studies have found that the gel property of tilapia surimi can be significantly improved by microbial fermentation [8]. However, most studies only focus on the improvement of flavor and gelation properties of tilapia surimi by fermentation with various strains of LAB, such as *Lactiplantibacillus plantarum* and *Pediococcus pentosaceus* [9,10], but ignore the formation of biogenic amines (BAs) during fermentation.

BAs in fermented foods are generally produced through the decarboxylation of free amino acids by microorganisms [11]. An excessive intake of BAs can result in severe harm to human health. Dietary exposure to foods containing high levels of histamine and tyramine is associated with many adverse health effects, including elevated blood pressure, migraines, and tachycardia [12], while the toxicity of cadaverine and putrescine in aquatic organisms has also been reported [13], indicating the importance to control BAs in foods. High concentrations of BAs have been found in fermented surimi because of the high free amino acids (FAAs) and active microbial metabolisms [8,14]. Natural microbial communities and even starters are involved in the production of BAs [14,15]. Therefore, it is important to select good starters to control the formation of BAs during fermentation. Many starters have been found to have a good BA reduction ability, such as the strains of LAB [16].

Therefore, in this study, in order to inhibit the production of BAs in the tilapia surimi, two strains of LAB were selected and used for collaborative fermentation. The changes in eight BAs during the collaborative fermentation process were studied and compared with those during the natural fermentation, followed by the study of FAAs and microbial community. The correlation networks in the group dimension were constructed to reveal the inhibition mechanism of BAs by collaborative fermentation. This work is expected to provide a useful way to enhance the food safety of fermented tilapia surimi.

## 2. Materials and Methods

### 2.1. Preparation of Starter Strains

The starter strains *Latilactobacillus sakei* H30-5 and *Pediococcus acidilactici* H30-21 were isolated from tilapia surimi in the natural fermentation process [2]. Both strains stored in the magnetic beads at −80 °C were anaerobically incubated into the de Man, Rogosa and Sharpe (MRS) medium (Guangdong Huankai Microbial Sci. & Tech. CO., Ltd., Guangzhou, China) at 30 °C for 24 h, respectively. After centrifugation at 4 °C and 10,000× *g* for 5 min, the precipitation was resuspended into 0.9% NaCl solution as starters.

### 2.2. Tilapia Surimi Fermentation

The tilapia surimi was fermented according to the previous study [17]. Briefly, the minced tilapia muscle was mixed with 0.5% sucrose, 0.5% glucose, 2% salt, and 5% water. The starter strains (1 × 10^6^ CFU/g) were added into the mixture. After blending for 5 min at ice temperature, the mixture was added into the plastic casings (Φ = 3 cm) and then fermented at 25 °C. The tilapia surimi fermented with starters at various time (18, 30, and 45 h) was taken as the CM group. The tilapia surimi fermented naturally was chosen as the CK group.

### 2.3. Detection of BAs

The tilapia surimi (5.0 g) was homogenized with 0.4 mol/L HClO_4_ solution. After centrifugation for 5 min at 10,000× *g* and 4 °C, the supernatant was taken to determine the content of BAs [8]. Briefly, the samples were first derived by dansyl chloride (Anpel Laboratory Technologies Inc., Shanghai, China), and were then detected through the LC-20AD HPLC system (Shimadzu, Kyoto, Japan) with the ChromCore™ C18A reversed-phase chromatographic column (250 × 4.6 mm; Nano Chrom, Suzhou, China) by an ultraviolet detector at 254 nm. The content of BAs was then calculated through the standard curves of 8 BAs (Sigma-Aldrich, St. Louis, MO, USA).

### 2.4. Detection of FAAs

The tilapia surimi (5.0 g) was first homogenized with 45 mL water. After centrifugation for 5 min at 10,000× *g* and 4 °C, the supernatant was taken to analyze the content of FAAs [8]. Briefly, the samples were derived by 9-fluorenylmethyl chloroformate (FMOC; Sigma-Aldrich, St. Louis, MO, USA) and o-phthalaldehyde (OPA; Sigma-Aldrich, St. Louis, MO, USA), respectively. The FAAs were then analyzed by the Agilent 1100 Series HPLC System (Agilent, Santa Clara, CA, USA) with the ZORBAX Eclipse AAA column (150 mm × 4.6 mm; Agilent, Santa Clara, CA, USA). The derived FAAs by FMOC were detected through a fluorescence detector (excitation at 266 nm and emission at 305 nm), while the derived FAAs by OPA were detected through an ultraviolet detector at 338 nm. The content of FAAs was calculated through the standard curves of 25 FAAs (Sigma-Aldrich, St. Louis, MO, USA).

### 2.5. Analysis of Microbial Community

The tilapia surimi (3.0 g) was first homogenized with 0.9% NaCl solution. The homogenized solution was then centrifuged for 5 min at 10,000× *g* and 4 °C. After genomic DNA extraction from the precipitation, the 16S rRNA gene high-throughput sequencing was used to analyze the microbial community according to the previous study [18]. Briefly, the V3-V4 regions of 16S rRNA genes were amplified by the PCR through the primer pair, 338F (5′-ACTCCTACGGGAGGCAGCAG-3′) and 806R (5′-GGACTACHVGGGTWTCTAAT-3′). After purification, the PCR amplicons were paired-end sequenced through the MiSeq PE300 platform (Illumina, San Diego, CA, USA). The operational taxonomic units (OTUs) were collected under 97% sequence similarity by UPARSE v7.0.1090. The taxonomy was analyzed by RDP Classifier v2.11 using the OTUs.

### 2.6. Statistical Analysis

Each experiment was performed in triplicate. The data were expressed as mean ± standard deviation. The content heatmap and correlation heatmap were constructed by MetaboAnalyst v6.0. The correlation network was drawn through Cytoscape v3.8.1. The influence of genus on the decrease in BA (*I_g_*) was calculated through the equation below according to the previous study [2] with some modification:$$I_{g}=\left|r\times A_{g}\right|$$
where *r* is the Pearson’s correlation coefficient between BA and genus (*p* < 0.05), and *A_g_* is the mean abundance of genus for Pearson’s correlation analysis.

## 3. Results and Discussion

### 3.1. Changes in BAs After Collaborative Fermentation

BAs are one of the most important food safety indexes in fermented foods. In this work, the changes in eight BAs during the fermentation process were studied in both the CM and CK groups (Figure 1 and Appendix A). Most of the BAs in both groups were obviously enhanced along with fermentation and maximized at 45 h. Histamine, cadaverine, putrescine, and tyramine were the main BAs in the fermented tilapia surimi. These BAs are also reported as the main BAs in fish sauce [18]. In contrast, tryptamine, phenylethylamine, spermine, and spermidine had relatively low contents (<10 mg/kg). As the most toxic BA, histamine is abundant in fermented food and is required to be limited to 50 mg/kg in aquatic products by the Food and Drug Administration of USA. In this study, less histamine was found in the tilapia surimi before fermentation (<1 mg/kg), but the histamine content after natural fermentation for 18, 30, and 45 h increased to 38.66, 44.18, and 39.09 mg/kg, respectively, with a risk of exceeding the standard. After collaborative fermentation, the histamine content was reduced by 25.84%, 33.59%, and 8.73% at 18, 30, and 45 h, respectively, suggesting good histamine inhibition by starters, especially at 30 h. As another toxic BA, tyramine was relatively low before natural fermentation for 30 h (<10 mg/kg), but was significantly elevated after natural fermentation for 45 h, reaching 20.99 mg/kg. Interestingly, the tyramine content obviously decreased to below 0.2 mg/kg during the whole collaborative fermentation process. Putrescine and cadaverine can not only evaluate the freshness of aquatic products, but also enhance the toxicity of histamine by protecting it from degradation [8]. Although a high content of putrescine was found after natural fermentation for 18–45 h (>10 mg/kg), its production is completely suppressed by collaborative fermentation with starters. There was no obvious change in the cadaverine content between the CK and CM groups before 30 h, but its content significantly decreased by 27.97% at 45 h after collaborative fermentation with starters. Similar to putrescine, the tryptamine is completely suppressed by collaborative fermentation with starters. There was no obvious change in the contents of phenylethylamine, spermine, and spermidine between the CM and CK groups. The total BA (TBA) significantly increased during the natural fermentation and reached the maximum (399.72 mg/kg) at 45 h. Interestingly, the TBA decreased by 42.37%, 35.86%, and 48.89% after collaborative fermentation with starters for 18, 30, and 45 h, respectively. The results suggested that the collaborative fermentation with starters could effectively inhibit the BA formation in the tilapia surimi.

### 3.2. Changes in FAAs After Collaborative Fermentation

As the raw material of BAs, the content of FAAs has great influence on the production of BAs. In this work, the changes in 25 FAAs during the fermentation process were studied in both the CM and CK groups (Figure 1 and Appendix A). Most FAAs in the CK group were obviously enhanced along with fermentation and maximized at the end of fermentation, while most FAAs in the CM group changed little in the fermentation process. Glycine and alanine were the most abundant FAAs and accounted for over 50% of the total FAAs, contributing to the sweet taste of tilapia surimi. This result was similar to other fermented freshwater fish surimi [14,19]. Compared with those in the CK group, most FAAs in the CM groups were reduced, suggesting a decrease in proteolytic activity by microorganisms. Interestingly, glutamic acid and aspartic acid were significantly elevated by collaborative fermentation, especially at the end of fermentation, resulting in an increase in the umami taste of tilapia surimi. In addition, the asparagine content was also significantly elevated in the CM group compared with that in the CK group, contributing to the increasing sweet taste of tilapia surimi. As the precusor of histamine, the histidine content significantly reduced with the increase in time in both groups, mainly due to the use in the production of high content of histamine. An obvious reduction in histidine was also observed in the previous study. Compared with those in the CK group, the contents of histidine, tyrosine, arginine, lysine, tryptophan, and phenylalanine were lower in the CM groups, contributing to the decrease in the corresponding BAs after collaborative fermentation. Similarly, the total FAA (TFAA) decreased by 11.84%, 12.66%, and 33.56% after collaborative fermentation with starters for 18, 30, and 45 h, respectively. The decrease in proteolytic activity after collaborative fermentation with starters was an important reason for the decrease in BAs.

### 3.3. Changes in Microbial Composition After Collaborative Fermentation

The metabolism of spoilage microorganisms is the main source of BAs. In this study, the changes in various microbial genera during the fermentation process were studied in both the CM and CK groups (Figure 2). A total of 15 genera (relative abundance > 0.5%) were detected. *Acinetobacter* (44.03%), *Micrococcus* (22.61%), and *Streptococcus* (14.15%) were the main genera in the raw tilapia surimi, in accordance with the previous result [8]. However, after natural fermentation, their abundance significantly decreased, while the abundance of LAB, including *Lactococcus* and *Latilactobacillus*, was markedly elevated, reaching 55.88% and 11.31% at the end of fermentation, respectively. In the CM group, besides *Acinetobacter*, *Micrococcus*, and *Streptococcus*, *Pediococcus* and *Latilactobacillus* were the dominant genera before fermentation due to the addition of starters. Similarly, along with the fermentation, spoilage microorganisms such as *Acinetobacter*, *Micrococcus*, and *Streptococcus* were remarkedly reduced after collaborative fermentation, and decreased to only 0.01%, 0.26%, and 1.56% at 45 h, respectively. The abundance of *Pediococcus* also obviously decreased, and its relative abundance was only 5.64% at 45 h, indicating the low adaptability of starter *P. acidilactici* to the tilapia surimi environment. In contrast, *Latilactobacillus* rapidly became the dominant genus after 18 h to 45 h of collaborative fermentation, with relative abundance > 50%. In addition, *Lactococcus* and *Enterobacter* were also abundant after collaborative fermentation, and their relative abundance reached 9.40% and 8.27% at the end of fermentation. It suggested that the surimi fermentation environment was more suitable for the growth of LAB, especially the starter *L. sakei*. A similar result has also been found in other studies [9,10].

### 3.4. Correlation of Core Microbial Genus with BAs and FAAs in Group Dimension

The microbial community and quality indexes of fermented food will change when the starter is added. The group-dimension correlation analysis can uncover the change mechanisms of various qualities induced by the microbial community [17,20]. However, most studies only perform the correlation analysis at the end of fermentation, resulting in the incomplete disclosure mechanisms [2,20]. In this study, the correlation analysis among 10 core microbial genera, 8 BAs, and 26 FAAs over different fermentation times between the CM and CK groups was performed through the Pearson’s correlation analysis (Appendix A), followed by the making of correlation networks (Figure 3).

In the 18 h period, a total of two genus clusters were found (Appendix A), including 18 h-1 (*Latilactobacillus*, *Pediococcus*, *Macrococcus*, and *Enterobacter*) and 18 h-2 (*Lactococcus*, *Acinetobacter*, *Vagococcus*, *Streptococcus*, *Morganella*, and *Aeromonas*). Most BAs and FAAs were positively correlated with the cluster 18 h-2, but were negatively related with the cluster 18 h-1. In the cluster 18 h-1, *Lactiplantibacillus*, *Pediococcus*, and *Enterobacter* showed a significantly negative correlation with histamine, tyramine, putrescine, tryptamine, phenylethylamine, and TBA (Figure 3A). The rise in their abundance after collaborative fermentation contributed to the decrease in BA content in the tilapia surimi at 18 h. In addition, *Lactiplantibacillus*, *Pediococcus*, and *Enterobacter* also showed a significantly negative correlation with most FAAs, especially TFAA (Figure 3A), suggesting their important role in the decrease in FAA content at 18 h.

Similarly, in the 30 h period, two genus clusters were observed (Appendix A), including 30 h-1 (*Latilactobacillus*, *Acinetobacter*, *Pediococcus*, *Enterobacter*, and *Aeromonas*) and 30 h-2 (*Lactococcus*, *Vagococcus*, *Streptococcus*, *Morganella*, and *Macrococcus*). Most BAs and FAAs exhibited a positive correlation with the cluster 30 h-2, but were negatively related with the cluster 30 h-1. In the cluster 30 h-1, *Lactiplantibacillus*, *Pediococcus*, *Enterobacter*, and *Aeromonas* exhibited a significantly negative correlation with histamine, tyramine, putrescine, tryptamine, and TBA (Figure 3B). The improvement in their abundance after collaborative fermentation was the main reason for the BA decrease at 30 h. In addition, *Lactiplantibacillus*, *Pediococcus*, *Enterobacter*, and *Aeromonas* were also significantly negatively correlated with most FAAs, especially TFAA (Figure 3B), suggesting their main role in the decrease in FAA content after collaborative fermentation for 30 h. In addition, the metabolism of these genera also led to the improvement in glutamic acid and aspartic acid at 30 h.

In the 45 h period, two genus clusters were also observed (Appendix A), including 45 h-1 (*Latilactobacillus*, *Acinetobacter*, *Pediococcus*, *Macrococcus*, and *Enterobacter*) and 45 h-2 (*Lactococcus*, *Vagococcus*, *Streptococcus*, *Morganella*, and *Aeromonas*). Most BAs and FAAs also showed a positive correlation with the cluster 45 h-2, but exhibited a negative correlation with the cluster 45 h-1. In cluster 45 h-1, *Lactiplantibacillus*, *Pediococcus*, *Macrococcus*, and *Enterobacter* exhibited a significantly negative correlation with histamine, tyramine, putrescine, cadaverine, tryptamine, phenylethylamine, and TBA (Figure 3C). The increase in their abundance after collaborative fermentation was the main reason for the BA decrease at 45 h. In addition, *Lactiplantibacillus*, *Pediococcus*, *Macrococcus*, and *Enterobacter* were also remarkedly negatively correlated with most FAAs, especially TFAA (Figure 3C), suggesting their main roles in the decrease in FAA content at 45 h. In addition, these genera also contributed to the increase in glutamic acid and aspartic acid at the end of fermentation after collaborative fermentation.

According to the correlation networks in different periods, *Lactococcus*, which possessed the highest mean abundance, showed a significantly positive correlation with most BAs in all three periods, suggesting its main role in the BA formation in the fermented tilapia surimi. *Lactococcus* is also found to show a correlation with BAs during the fermentation of different foods [8,21]. Many strains in the genus *Lactococcus* have been reported to produce BAs in the cheese [22]. Meanwhile, many spoilage microorganisms, especially *Vagococcus* and *Streptococcus*, were also responsible for the BA formation in the fermented tilapia surimi in the three periods. Similarly, *Vagococcus* is found to be related to the formation of BAs in vacuum-packaged large yellow croaker during ice storage [23]. Many strains in the genus *Streptococcus* isolated from homemade natural yogurt have been reported to produce BAs [24]. In this study, the decrease in these genera might be the main reason for the decrease in BAs after cooperative fermentation with starters. In addition, the strains in *Lactococcus* are famous for their proteolytic activity [25,26,27], followed by the strains in *Streptococcus* [28,29,30]. The inhibition of their proteolytic activity after cooperative fermentation with starters also contributed to the decrease in BAs in the fermented tilapia surimi.

During the tilapia surimi fermentation in different periods, *Lactiplantibacillus*, *Pediococcus*, *Macrococcus*, *Enterobacter*, and *Aeromonas* played important roles in the decrease in BAs in different periods. The microorganisms in fermented food that are high in abundance and can obviously affect the food quality are considered the core ones [20,31]. In this work, after considering both factors, the influence of the genus was further calculated (Figure 4). According to the influence of these genera, *Latilactobacillus* contributed the most to the decrease in BAs, with a contribution over 71%, 73%, and 76% in the 18 h, 30 h, and 45 h periods, respectively. It suggested that the starter *L. sakei* H30-5 contributed the most to the decrease in BAs in the tilapia surimi. The acidification of LAB has been found to suppress the growth of spoilage microorganisms and their metabolisms [32], resulting in the inhibition of production of BAs. Similar results have reported that *L. sakei* can improve the food safety of fermented meat sausage [33,34]. Moreover, in this study, the high abundance of the starter *P. acidilactici* at the beginning of inoculation could quickly reduce the acidity of the fermentation environment, which was ideal for the starter *L. sakei* to grow and quickly reach a dominant position. The collaborative fermentation of these two starters kept the fermentation environment at a low pH and consequentially kept spoilage microorganisms at a low abundance. The collaborative fermentation of starters have also been reported to possess more advantages than a single starter in the formation of gel strength and flavor [2,17]. In addition, the inhibition of spoilage microorganisms by starters also led to the decrease in precusor FAAs of BAs, contributing to the decrease in BAs in the tilapia surimi. The collaborative metabolism of starters was useful in controlling the production of BAs from spoilage microorganisms.

## 4. Conclusions

Most BAs in both the CK and CM groups were obviously enhanced during fermentation. Histamine, cadaverine, putrescine, and tyramine were the main BAs. Most BAs and the TBA were significantly inhibited through the collaborative fermentation with starters. Most FAAs, especially the precusors of the BAs, as well as TFAA, were significantly reduced, while the umami FAAs, including glutamic acid and aspartic acid, were significantly enhanced through the collaborative fermentation with starters. The spoilage microorganisms such as *Acinetobacter*, *Micrococcus*, and *Streptococcus*, as well as *Pediococcus*, were significantly inhibited after collaborative fermentation, while *Latilactobacillus* rapidly rose to the dominant abundance, suggesting the better environment adaptability of *L. sakei* compared to *P. acidilactici*. The group-dimension correlation network found that *Lactiplantibacillus*, *Pediococcus*, *Macrococcus*, *Enterobacter*, and *Aeromonas* played important roles in the decrease in BAs in different periods, especially *Latilactobacillus*, which had the greatest influence on the decrease in BAs. The acidification of LAB. especially the starter *L. sakei*, could suppress the growth of spoilage microorganisms and their metabolisms, resulting in the inhibition of the production of BAs. Collaborative fermentation with starters was first used to control the BAs in the tilapia surimi and exhibited good performance. *L. sakei* and *P. acidilactici* can be used as the special starters to enhance the food safety of fermented tilapia surimi in the future.

## Figures and Tables

**Figure 1 foods-13-03297-f001:**
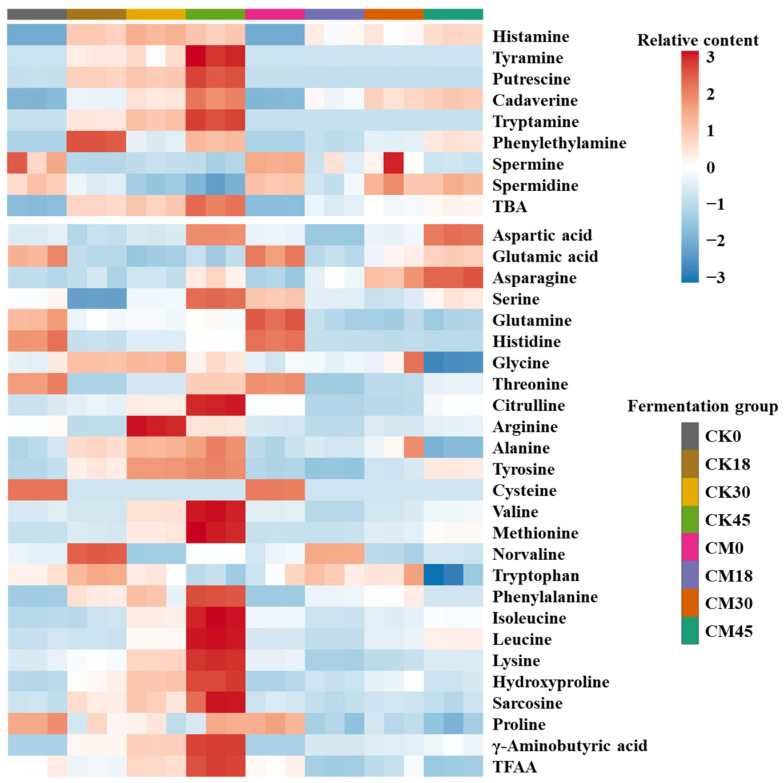
Changes in BAs and FAAs in tilapia surimi during the fermentation process of CM and CK groups.

**Figure 2 foods-13-03297-f002:**
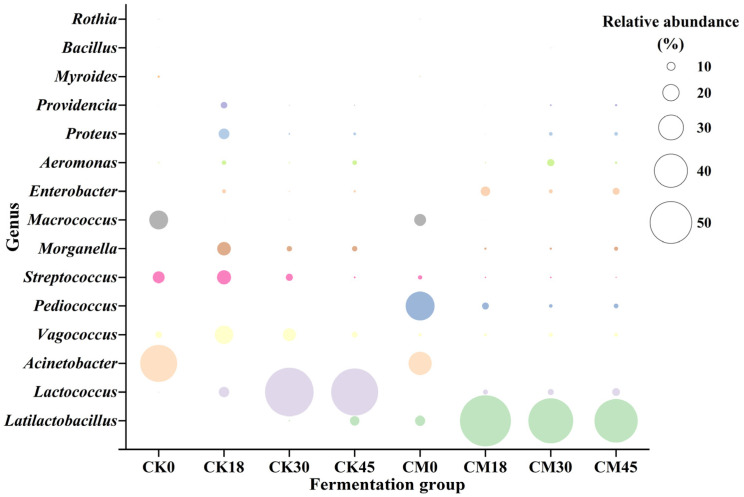
Changes in microbial taxonomic compositions in the tilapia surimi during the fermentation process of the CM and CK groups.

**Figure 3 foods-13-03297-f003:**
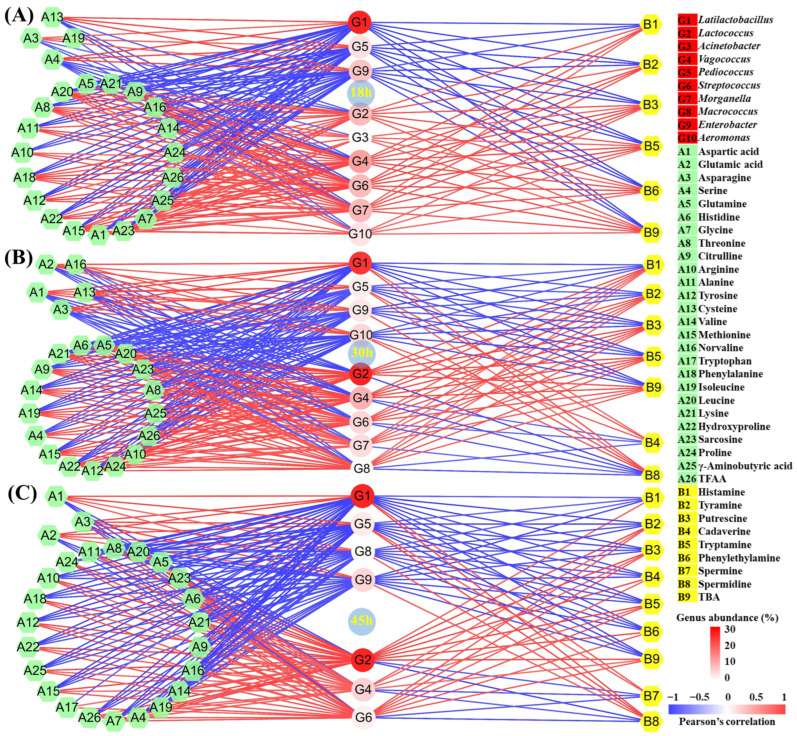
Correlation network constructed by the Pearson’s correlation coefficient among the core microbial genera, BAs, and FAAs in the tilapia surimi in the group dimension in the (**A**) 18 h, (**B**) 30 h, and (**C**) 45 h periods. The red line and blue line, respectively, represent the significantly positive and negative correlation (|r| > 0.8 and *p* < 0.05).

**Figure 4 foods-13-03297-f004:**
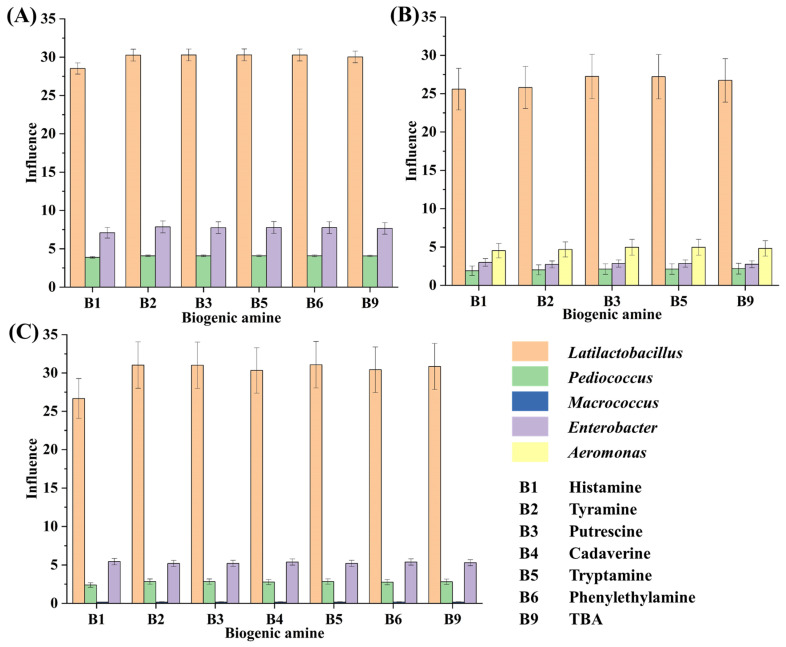
The influence of the genus on the decrease in BAs in the tilapia surimi in the (**A**) 18 h, (**B**) 30 h, and (**C**) 45 h periods.

## Data Availability

The original contributions presented in the study are included in the article and Appendix A, further inquiries can be directed to the corresponding authors.

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
