# Peer review of "Inhibition of Biogenic Amines in Fermented Tilapia Surimi by Collaborative Fermentation of *Latilactobacillus sakei* and *Pediococcus acidilactici"

_foods, 2024, doi:10.3390/foods13203297_

Round 1
Reviewer 1 Report
Comments and Suggestions for Authors
Comments for authors
The paper entitled “Inhibition of Biogenic Amines in Fermented Tilapia Surimi by 2 Collaborative Fermentation of Latilactobacillus sakei and Pediococcus acidilactici” is very interesting research; however, it is necessary to adjust throughout the text:
1. Line 44-46. You indicate that there are studies with various strains, however, in the reference documents 4 are mentioned (Lactiplantibacillus plantarum H30-2, Pediococcus acidilactici H30-21, Latilactobacillus sakei and Pediococcus acidilactici), I suggest you indicate the most reported strains to avoid adjectives that do not give much information.
2. At the end of the introduction you should adequately state the objective of the work.The references are mostly current, I only recommend you to update the ones that are from 2004 to 2013, most of them do not allow the article to be visualized as a current topic (frontier of science).
3. The introduction is very terse about the importance of inhibiting the production of biogenic amines, it should highlight the importance of this phenomenon as the work is fully supported by this.
4. In materials and methods, you indicate that you isolated two strains and you assure that they are Latilactobacillus sakei H30-5 and Pediococcus acidilactici H30-21, how do you know exactly what these strains are, you should give details of this as it is of extreme importance in your document. Did you perform a molecular identification, how did you characterize them?
5. line 73 and 81, should be expressed in g when referring to centrifugation, since this is the best way to reproduce
6. Sections 2.1-2.4 lack bibliographical support.
7. You indicate that the results of collaborative fermentation suggest that it inhibits the formation of biogenic amines, it would be very relevant to use the reports on the phenomenon to discuss on a scientific basis why it is possible that collaborative fermentation inhibits the formation of biogenic amines.
8. Figure 1 should be explained clearly and with more emphasis. Is this figure necessary in the text?
9. The presence of FAAs may be due to a higher expression of proteolytic activity, clarify and complete section 3.2 and 3.4.
10. In figure 4 you must add error bars
11. In the conclusions it should be more forceful to make clear what the contribution of the work is. What is your recommendation or suggestion?
Author Response
Dear Reviewer 1:
Thanks for your letter and comments concerning our manuscript entitled “Inhibition of Biogenic Amines in Fermented Tilapia Surimi by Collaborative Fermentation of Latilactobacillus sakei and Pediococcus acidilactici” (foods-3218829).
Those comments are very valuable and helpful for revising and improving our paper. We have studied the comments carefully and have made correction which we hope to meet with approval. Revised portions are highlighted in yellow in the manuscript. The responds to the comments of reviewer 1 are as followings:
Comments to the author:
The paper entitled “Inhibition of Biogenic Amines in Fermented Tilapia Surimi by 2 Collaborative Fermentation of Latilactobacillus sakei and Pediococcus acidilactici” is very interesting research; however, it is necessary to adjust throughout the text:
Question 1:
Line 44-46. You indicate that there are studies with various strains, however, in the reference documents 4 are mentioned (Lactiplantibacillus plantarum H30-2, Pediococcus acidilactici H30-21, Latilactobacillus sakei and Pediococcus acidilactici), I suggest you indicate the most reported strains to avoid adjectives that do not give much information.
Answer 1:
Lactiplantibacillus plantarum and Pediococcus pentosaceus are the most reported strains, which have been added in the manuscript. See Line 45-46.
Question 2:
At the end of the introduction you should adequately state the objective of the work. The references are mostly current, I only recommend you to update the ones that are from 2004 to 2013, most of them do not allow the article to be visualized as a current topic (frontier of science).
Answer 2:
“In this study, the collaborative fermentation of two strains of LAB was used to inhibit BAs” has been changed to “Therefore, in this study, in order to inhibit the production of BAs in the tilapia surimi, two strains of LAB was selected and used for collaborative fermentation. The changes of 8 BAs during the collaborative fermentation process were studied and compared with those in the natural fermentation, followed by the study of free amino acids (FAAs) and microbial Community”. See Line 60-64.
The references from 2004 to 2013 have been updated according to the suggestions.
Question 3:
The introduction is very terse about the importance of inhibiting the production of biogenic amines, it should highlight the importance of this phenomenon as the work is fully supported by this.
Answer 3:
The sentences “Dietary exposure to foods containing high levels of histamine and tyramine is associated with many adverse health effects, including elevated blood pressure, migraines, and tachycardia [12], while the toxicity of cadaverine and putrescine in aquatic organisms has also been reported [13], indicating the importance to control BAs in foods. High concentrations of BAs have been found in fermented surimi because of the high free amino acids (FAAs) and active microbial metabolisms [8,14]. Natural microbial community and even starters are involved in the production of BAs [14,15]. Therefore, it is important to select good starters to control the formation of BAs during fermentation. Many starters have been found to have good ability of BA reduction, such as the strains of LAB [16]” have been added in the manuscript. See Line 50-59.
Question 4:
In materials and methods, you indicate that you isolated two strains and you assure that they are Latilactobacillus sakei H30-5 and Pediococcus acidilactici H30-21, how do you know exactly what these strains are, you should give details of this as it is of extreme importance in your document. Did you perform a molecular identification, how did you characterize them?
Answer 4:
Latilactobacillus sakei H30-5 and Pediococcus acidilactici H30-2 were isolated from tilapia sausage during natural fermentation. The isolation method and identification result have been described in the previous study. The reference [2] has been added in the manuscript. See Line 70.
Question 5:
line 73 and 81, should be expressed in g when referring to centrifugation, since this is the best way to reproduce
Answer 5:
“12000 r/min” has been changed to “10000×g”. See Line 73, 84, 92, and 102.
Question 6:
Sections 2.1-2.4 lack bibliographical support.
Answer 6:
The references have been added in the Sections 2.1-2.4.
Question 7:
You indicate that the results of collaborative fermentation suggest that it inhibits the formation of biogenic amines, it would be very relevant to use the reports on the phenomenon to discuss on a scientific basis why it is possible that collaborative fermentation inhibits the formation of biogenic amines.
Answer 7:
The sentences “The acidification of LAB has been found to suppress the growth of spoilage microorgan-isms and their metabolisms [32], resulting in the inhibition of production of BAs. Similar results have been reported that L. sakei can improve the food safety of fermented meat sausage [33,34]. Moreover, in this study, the high abundance of starter P. acidilactici at the beginning of inoculation could quickly reduce the acidity of fermentation environment, which was ideal for the starter L. sakei to grow and quickly reached a dominant position. The collaborative fermentation of these two starters kept the fermentation environment at a low pH and consequentially kept spoilage microorganisms at a low abundance” can explain the inhibition mechanism of biogenic amines by collaborative fermentation. See Line 282-290.
Question 8:
Figure 1 should be explained clearly and with more emphasis. Is this figure necessary in the text?
Answer 8:
The Figure 1 can visually show the changes of BAs and FAAs in different fermentation groups. The data of BAs and FAAs have been supplemented in the Table S1 and Table S2.
Question 9:
The presence of FAAs may be due to a higher expression of proteolytic activity, clarify and complete section 3.2 and 3.4.
Answer 9:
The information related to proteolytic activity has been added in the section 3.2 and 3.4 of manuscript. See Line 165-166, 177-179, and 267-270.
Question 10:
In figure 4 you must add error bars
Answer 10:
The error bars have been added in the Figure 4.
Question 11:
In the conclusions it should be more forceful to make clear what the contribution of the work is. What is your recommendation or suggestion?
Answer 11:
The sentences “The collaborative fermentation with starters was first used to control the BAs in the tilapia surimi and exhibited good performance. L. sakei and P. acidilactici can be used as the special starters to enhance the food safety of fermented tilapia surimi in the future” have been supplemented in the Conclusions. See Line 311-314.
We earnestly appreciate for your warm work, and hope that these corrections will meet with approval.
Once again, thank you very much for your comments and suggestions.
Yours sincerely,
Yongqiang Zhao
E-mail: zhaoyq@scsfri.ac.cn
Reviewer 2 Report
Comments and Suggestions for Authors
The article entitled ''Inhibition of Biogenic Amines in Fermented Tilapia Surimi by Collaborative Fermentation of Latilactobacillus sakei and Pediococcus acidilactici'' is interesting and proposes an important topic, namely the formation of biogenic amines, compounds resulting from the decarboxylation of amino acids in processed products. There are some issues that need to be discussed before the article is published:
- The authors say that ''fermentation is a new method.'' I disagree with the use of this word, replace, line 14.
- the abstract does not end with a summary conclusion;
- the introduction does not contain the detailed aim of the study;
- not all abbreviations are explained; just a few examples: lines 61, 92, 94, 97, 98
- the spoilage microorganism source is not specified or discussed clearly, please develop a little bit.
- cooperative fermentation, as keyword, is not described or debated enough in the text
- the conclusions part does not contain a main conclusion nor recommendations.
Author Response
Dear Reviewer 2:
Thanks for your letter and comments concerning our manuscript entitled “Inhibition of Biogenic Amines in Fermented Tilapia Surimi by Collaborative Fermentation of Latilactobacillus sakei and Pediococcus acidilactici” (foods-3218829).
Those comments are very valuable and helpful for revising and improving our paper. We have studied the comments carefully and have made correction which we hope to meet with approval. Revised portions are highlighted in yellow in the manuscript. The responds to the comments of reviewer 2 are as followings:
Comments to the author:
The article entitled ''Inhibition of Biogenic Amines in Fermented Tilapia Surimi by Collaborative Fermentation of Latilactobacillus sakei and Pediococcus acidilactici'' is interesting and proposes an important topic, namely the formation of biogenic amines, compounds resulting from the decarboxylation of amino acids in processed products. There are some issues that need to be discussed before the article is published:
Question 1:
The authors say that ''fermentation is a new method.'' I disagree with the use of this word, replace, line 14.
Answer 1:
“Fermentation is a novel method” has been changed to “Fermentation is an effective method”. See Line 13 and 41.
Question 2:
The abstract does not end with a summary conclusion;
Answer 2:
The sentence “L. sakei and P. acidilactici can be developed as the special starters to control the BA production in fermented tilapia surimi through collaborative fermentation” has been added at the end of Abstract. See Line 25-27.
Question 3:
The introduction does not contain the detailed aim of the study;
Answer 3:
The detailed aim of the study has been added in the introduction. See Line 60-61.
Question 4:
Not all abbreviations are explained; just a few examples: lines 61, 92, 94, 97, 98
Answer 4:
The abbreviations have been explained according to the suggestions. See Line 55, 71-72, and 108-108.
Question 5:
The spoilage microorganism source is not specified or discussed clearly, please develop a little bit.
Answer 5:
The spoilage microorganism source is mainly from the environment in which the tilapia surimi is made. The paragraph “According to the correlation networks in different periods, Lactococcus that possessed the highest mean abundance showed significantly positive correlation with most BAs in all three periods, suggesting its main role in the BA formation in the fermented tilapia su-rimi. Lactococcus is also found to show correlation with BAs during the fermentation of different foods [8,21]. Many strains in the genus Lactococcus have been reported to produce BAs in the cheese [22]. Meanwhile, many spoilage microorganisms, especially Vagococcus and Streptococcus, were also responsible for the BA formation in the fermented tilapia su-rimi in the three periods. Similarly, Vagococcus is found to be related to the formation of BAs in vacuum-packaged large yellow croaker during ice storage [23]. Many strains in the genus Streptococcus isolated from home-made natural yogurt have been reported to pro-duce BAs [24]. In this study, the decrease of these genera might be the main reason for the decrease of BAs after cooperative fermentation with starters. In addition, the strains in Lactococcus are famous for their proteolytic activity [25–27], followed by the strains in Streptococcus [28–30]. The inhibition of their proteolytic activity after cooperative fermenta-tion with starters also contributed to the decrease of BAs in the fermented tilapia surimi” has been added in the manuscript to discuss their relationship with BA formation. See Line 256-270.
Question 6:
Cooperative fermentation, as keyword, is not described or debated enough in the text.
Answer 6:
Thanks for the suggestions. The sentences “Moreover, in this study, the high abundance of starter P. acidilactici at the beginning of inoculation could quickly reduce the acidity of fermentation environment, which was ideal for the starter L. sakei to grow and quickly reached a dominant position. The collaborative fermentation of these two starters kept the fermentation environment at a low pH and consequentially kept spoilage microorganisms at a low abundance. The collaborative fermentation of starters have also been reported to possess more advantages than single starter in the formation of gel strength and flavor [2,17]” have been added in the manuscript. See Line 285-292.
Question 7:
The conclusions part does not contain a main conclusion nor recommendations.
Answer 7:
The sentences “The collaborative fermentation with starters was first used to control the BAs in the tilapia surimi and exhibited good performance. L. sakei and P. acidilactici can be used as the special starters to enhance the food safety of fermented tilapia surimi in the future” have been added in the Conclusions. See Line 311-314.
We earnestly appreciate for your warm work, and hope that these corrections will meet with approval.
Once again, thank you very much for your comments and suggestions.
Yours sincerely,
Yongqiang Zhao
E-mail: zhaoyq@scsfri.ac.cn